# Dual-Responsive “Egg-Box” Shaped Microgel Beads Based on W_1_/O/W_2_ Double Emulsions for Colon-Targeted Delivery of Synbiotics

**DOI:** 10.3390/foods13142163

**Published:** 2024-07-09

**Authors:** Xian He, Yunyun Qin, Haoyue Liu, Kang Cheng, Wanshui Yang, Xinsheng Qin

**Affiliations:** 1Department of Nutrition and Food Hygiene, Center for Big Data and Population Health of IHM, School of Public Health, Anhui Medical University, Hefei 230032, China; hx000312@163.com (X.H.); qinyunyun2023@163.com (Y.Q.); lhy12342024@163.com (H.L.); ck1312744283@163.com (K.C.); wanshuiyang@gmail.com (W.Y.); 2First Clinical Medical College, Anhui Medical University, Hefei 230032, China

**Keywords:** zein–apple pectin hybrid nanoparticles, W_1_/O/W_2_ emulsions, synbiotics, “egg-box” shaped microgel beads, dual-responsive

## Abstract

In this study, for enhancing the resistance of probiotics to environmental factors, we designed a microgel beads delivery system loaded with synbiotics. Multiple droplets of W_1_/O/W_2_ emulsions stabilized with zein–apple pectin hybrid nanoparticles (ZAHPs) acted as the inner “egg,” whereas a three-dimensional network of poly-L-lysine (PLL)-alginate-CaCl_2_ (Ca) crosslinked gel layers served as the outermost “box.” ZAHPs with a mass ratio of 2:1 zein-to-apple pectin showed excellent wettability (three-phase contact angle = 89.88°). The results of the ζ-potentials and Fourier transform infrared spectroscopy demonstrate that electrostatic interaction forces and hydrogen bonding were the main forces involved in the formation of ZAHPs. On this basis, we prepared W_1_/O/W_2_ emulsions with other preparation parameters and observed their microstructures by optical microscopy and confocal laser scanning microscope. The multi-chambered structures of W_1_/O/W_2_ emulsions were successfully visualized. Finally, the W_1_/O/W_2_ emulsions were coated with PLL-alginate-Ca using the solution extrusion method. The results of the in vitro colonic digestion stage reveal that the survival rate of probiotics in the microgel beads was about 75.11%, which was significantly higher than that of the free. Moreover, probiotics encapsulated in microgel beads also showed positive storage stability. Apple pectin would serve as both an emulsifier and a prebiotic. Thus, the results indicate that the “egg-box” shaped microgel beads, designed on the basis of pH-sensitive and enzyme-triggered mechanisms, can enhance the efficiency of probiotics translocation in the digestive tract and mediate spatiotemporal controlled release.

## 1. Introduction

Probiotics have been widely used in food and biomedical products and have offered tremendous functional benefits. The network mechanism of “probiotics–gut homeostasis–health” interactions has been consistently demonstrated in recent years. However, the amounts of probiotics that are utilized should be no less than the minimum effective dose (10^6^ CFU/g) required to provide health benefits [1,2]. Microencapsulation technology has been widely studied for the protection of probiotics thanks to the advantage it offers in not requiring the use of extreme stress treatments, including extrusion, emulsification, microfluidization, etc. [3]. It is worth mentioning that W_1_/O/W_2_ emulsions are favored by researchers owing to their special multi-chambered structure. The hydrophilic bioactives are encapsulated in the inner aqueous phase (W_1_ phase) to prevent their exposure to the harmful elements. Our previous study found that W_1_/O/W_2_ emulsions stabilized with whey protein isolate/(-)-epigallocatechin-3-gallate covalent conjugate nanoparticles improve the survival of *Lactobacillus plantarum* in the digestive tract. *Lactobacillus plantarum* was encapsulated in the W_1_ phase. The outer aqueous phase (W_2_ phase) and the intermediate oil phase (O phase) provide multi-layered physical barriers for probiotics, allowing them to avoid direct contact with H^+^, digestive enzymes and bile salts [4]. In order to expand the commercial applications of emulsion delivery systems for encapsulated probiotics in the food and pharmaceutical sectors, researchers have shown great interest in developing green, biocompatible stabilizers (e.g., protein–polysaccharide polymers) [5,6]. In this study, zein and apple pectin were chosen as external phase (O/W_2_ phase) stabilizers for W_1_/O/W_2_ emulsions.

Zein nanoparticles with hydrophobic properties are resistant to digestive enzymes and allow effective prolongation of the storage time of food functional factors and drug molecules in the digestive tract for controlled release [7,8]. In addition, we will add apple pectin to form hybrid particles by electrostatic deposition. This effectively avoids low interfacial coverage and agglomeration of emulsions stabilized by zein due to hydrophobic attraction [9,10]. Apple pectin extracted from plant cell walls is a dietary fiber with gelling, thickening, immunomodulatory and anti-digestive properties [11]. It improves the interfacial stability of coated nanoparticles by decreasing hydrophobic attraction and increasing spatial and electrostatic repulsion [12]. Huang et al. have found that anionic pectin adsorbed on the surface of zein nanoparticles forms a shell layer under mildly acidic conditions (3.0 < pH < 6.5) [9]. At pH 7.0, the interface profile begins to blur. Strongly negatively charged zein nanoparticles and pectin repel each other under electrostatic forces, leading to the desorption of pectin from the surface of the encapsulated zein nanoparticles. The pH values mentioned above largely coincide with the changes in pH in the various stages of the human gastrointestinal tract. Thus, zein–pectin particles are probably promising materials for designing pH-responsive stimulators. Furthermore, apple pectin, as a prebiotic, can be broken down by specific enzymes in the colon to produce butyric acid and improve dysbiosis [13,14]. Our previous study also showed that the W_1_/O/W_2_ emulsion delivery system of co-encapsulated prebiotics and probiotics can offer favorable synergistic effects [15]. In summary, zein–apple pectin hybrid particles seem to be excellent pH-responsive and enzyme-triggered materials.

However, W_1_/O/W_2_ emulsions are susceptible to flocculation, coalescence and Ostwald ripening under the influence of mechanical stress, thermal energy, time and pH [15,16]. For the ultimate goal of achieving controlled colon-targeted release of synbiotics to improve the bioavailability of probiotics, we devise a dual-responsive delivery system. Firstly, multiple droplets of W_1_/O/W_2_ emulsions stabilized by zein–apple pectin hybrid particles (ZAHPs) act as the inner “egg.” In this study, poly-L-lysine (PLL) will be added to W_1_/O/W_2_ emulsions for preliminary electrostatic coating [17]. The addition of PLL could effectively reduce the porosity of calcium alginate and form a “box” structure of dense network gel with superior barrier properties. At first, ZAHPs with different mass ratios were prepared as emulsifiers for the O/W_2_ phase of W_1_/O/W_2_ emulsions by the antisolvent precipitation method. They were characterized by particle size, ζ-potential, Fourier transform infrared spectroscopy (FTIR), surface contact angularity meter and scanning electron microscopy (SEM). Polyglycerol polyricinoleate (PGPR) was selected as the edible surfactant for the W_1_/O phase. The microstructures of the W_1_/O/W_2_ emulsions were preliminarily observed by optical microscopy and confocal laser scanning microscopy (CLSM). Then, the physical properties of W_1_/O/W_2_ emulsions were further explored (e.g., mechanical properties, centrifugal stability and storage stability). In this study, “egg-box” shaped microgel beads were fabricated by solution extrusion method. Ultimately, differences in storage stabilities and gastrointestinal digestive resistances of “egg-box” shaped microgel beads loaded with synbiotics were evaluated. The microscopic and macroscopic morphologies of the microgel beads at various stages of digestion were documented by an optical microscopy and a camera.

## 2. Materials and Methods

### 2.1. Materials

Zein (purity ≥ 92%), fructooligosaccharides (FOS), poly-L-lysine (PLL), bile salts, sodium alginate, polyglycerol polyricinoleate (PGPR) were acquired from Shanghai Yuanye Bio-Technology Co., Ltd. (Shanghai, China). Apple pectin (AP, galacturonic acid content above 65%) was provided by Beijing Jinming Biotechnology Co., Ltd. (Beijing, China). Nile red, CaCl_2_ and corn oil were provided by Shanghai Aladdin Biochemical Technology Co., Ltd. (Shanghai, China). Nile blue A, pepsin (P7000, ≥250 U/mg), pancreatic enzymes (P7545, 8× USP specifications) were sourced from Sigma-Aldrich Co. (St. Louis, MO, USA). *Lactobacillus reuteri* strains were purchased from CICC Co., Ltd. (Beijing, China) Medium DSMZ 11 was acquired from Shandong Tuopu Biol-engineering Co., Ltd. (Zhaoyuan, China). Anaerobic gas production packages were purchased by MITSUBISHI GAS CHEMICAL (Tokyo, Japan). All other chemicals used in the experiments were of analytical grade. The glass instruments were all sterilized.

### 2.2. Fabrication of Zein–Apple Pectin Hybrid Nanoparticles (ZAHPs)

Anti-solvent precipitation method was used to prepare ZAHPs with minor modifications [18]. In short, different masses of zein were dissolved in an aqueous ethanol solution (70%, *v*/*v*) with magnetic stirring. Next, the solution was dropped into 120 mL deionized water by stirring for 30 min. At the same time, powdered AP was dissolved in deionized water and stirred for 3 h. The two dispersions mentioned above were mixed and homogenized by a high-speed homogenizer (T25 Ultra Turrax, Staufen im Breisgau, Germany) at 7000 rpm for 6 min. The ethanol-free solutions of zein and AP were obtained by rotary evaporator at 45 °C. Finally, the mass ratios of zein-to-AP in the mixed solutions were 10:0, 8:1, 4:1, 2:1, 1:1 and 0:10. The pH of the solutions was adjusted to 4.0 with 1 M HCl.

### 2.3. Characterization of ZAHPs

#### 2.3.1. Particle Size, Polydispersity Index (PDI), and Zeta (ζ)-Potential

The particle size, PDI, and ζ-Potential of zein, AP, and ZAHPs were determined using the Zetasizer Nano 2000 (Malvern Instruments Ltd., Worcestershire, UK) with a dispersion coefficient of 1.33 and an equilibrium time of 120 s. The particles were diluted 20 times prior to analysis.

#### 2.3.2. Fourier Transform Infrared (FTIR) Spectroscopy

The interaction of the particles was characterized by Fourier transform infrared spectroscopy (Nicolet iS10, Thermo Fisher, Waltham, MA, USA). An amount of 200 mg KBr was added to a 2 mg lyophilized powder of zein, AP and ZAHPs. Spectra were obtained after the samples were made into cylindrical tablets by a tablet press.

#### 2.3.3. Three-Phase Contact Angle (θ)

The *θ* values of particles were recorded via a contact angle meter (OCA Pro15 Dataphysics, Stuttgart, Germany). A series of 0.1 g lyophilized powders of zein, AP, and ZAHPs were pressed into cylindrical tablets and immersed into the square cuvettes containing purified corn oil. Using a high-precision syringe, 3 μL of deionized water was gently extended into the oil droplet and slowly dripped onto the sample sheet, taking care to exhaust air bubbles in the syringe. The *θ* of the samples were recorded by a high-speed camera.

#### 2.3.4. Scanning Electron Microscopy (SEM)

The microstructures of the particles were surveyed using a scanning electron microscope (ZEISS Gemini SEM 300, Oberkochen, Germany). Before the experiment, 10 mg of lyophilized zein, AP, and ZAHP powders were applied to the metal conductive adhesive on the carrier stage, and the gold layer was sprayed on double-sided adhesive sample via an ion sputtering apparatus (Cressington108 Auto, liverpool, UK). Then, the samples were examined at an acceleration voltage of 3.0 kV.

### 2.4. Fabrication of W_1_/O/W_2_ Emulsions without Synbiotics

W_1_/O/W_2_ emulsions were prepared by a two-step homogenization method [19]. (i) Preparation of the dispersed phase (W_1_/O phase): Different PGPR concentrations (0.5%, 1%, 2%, 3%, 4%, *w*/*v*) were dissolved in corn oil at 40 °C as surfactants. Next, the O phase was slowly stirred for 10 min. After the O phase was cooled to room temperature, the W_1_ phase was slowly poured into the O phase and homogenized at 10,000 rpm for 3 min. (ii) Fabrication of W_1_/O/W_2_ emulsions: A 3% ZAHP in the W_2_ phase was used as emulsifier. The immiscible dispersions were homogenized under stirring at 5000 rpm for 2 min to fabricate 5 30 mL W_1_/O/W_2_ emulsions with different volume ratios of W_1_/O phase-to-W_2_ phase (4:6, 5:5, 6:4, 7:3 and 8:2). The samples were refrigerated for future use.

### 2.5. Characterization of W_1_/O/W_2_ Emulsions

#### 2.5.1. Optical Microstructures and Visible Graphics of W_1_/O/W_2_ Emulsions

The optical microstructures of W_1_/O/W_2_ emulsions were characterized via a fluorescence microscope (ZEISS Axio Vert.A1, Oberkochen, Germany) with a magnification of 50×, 100×, and 200×. The diluted W_1_/O/W_2_ emulsions were slowly dropped on a slide and observed at room temperature. The visible pictures of these were recorded by a camera.

#### 2.5.2. Confocal Laser Scanning Microscope (CLSM)

Microstructures of fluorescent coloring of W_1_/O/W_2_ emulsions were characterized by a confocal laser scanning microscope (ZEISS LSM880 + airyscan, Oberkochen, Germany). The samples were marked with Nile Red and Nile Blue A. After 12 h of storage away from light, the microstructures of droplets were observed at a scanning frequency of 1024 × 1024 and under a 40× oil microscope. The laser wavelengths were 488 nm (Nile Red) and 633 nm (Nile Blue A).

#### 2.5.3. Rheology

The rheological properties of W_1_/O/W_2_ emulsions were determined by a rheometer (Anton Paar MCR 302e, Graz, Austrian). Before the experiment, W_1_/O/W_2_ emulsions were allowed to equilibrate on the plate for 5 min. The apparent viscosity curves and frequency sweep curves of W_1_/O/W_2_ emulsions fabricated with different preparation parameters were determined separately.

### 2.6. Stability Analysis of W_1_/O/W_2_ Emulsions

#### 2.6.1. Storage Stability and Creaming Index (CI) of W_1_/O/W_2_ Emulsions

The storage stabilities of the W_1_/O/W_2_ emulsions were indicated using visual photographs and CI at 4 °C.
Creaming index (%) =H_t_/H_e_ × 100% 

From this equation, H_e_ refers to the total heights of the initial emulsions (cm) and H_t_ refers to the heights of the retained emulsions observed during the storage period (cm).

#### 2.6.2. Centrifugation Stability

The mechanical resistances of the W_1_/O/W_2_ emulsions were determined by centrifugation. An amount of 1 mL of freshly prepared W_1_/O/W_2_ emulsions was centrifuged at 3000 rpm for 2 min. The emulsification or the oil leakage that occurred with these emulsions was documented by a camera.

### 2.7. Configuration of “Egg-Box” Shaped Microgel Beads

#### 2.7.1. Activation and Cultivation of Probiotics

The lyophilized *Lactobacillus reuteri* powders were activated for 2–3 generations under anaerobic conditions. Next, the optical density values of the strain at 600 nm (OD_600_) were measured every 2 h and the growth curve of the strains for 24 h was plotted to determine the optimal growth time. Activated *Lactobacillus reuteri* were frozen and stored in glycerol at −20 °C. Prior to the experiment, *Lactobacillus reuteri* was cultured on agar plates to the desired concentration. The results were expressed in log CFU/g.

#### 2.7.2. Fabrication of “Egg-Box” Shaped Microgel Beads

The W_1_ phase was prepared by thorough mixing of probiotics (×10^10^ CFU/mL) and oligofructose (2%, *w*/*v*). The remaining steps were followed as per Section 2.4, above. An amount of 5 mL W_1_/O/W_2_ emulsions were crosslinked with PLL (0.1%, *w*/*v*) at 150 rpm. Then, 2 mL of 2% sodium alginate was added for further moderate mixing. Finally, microgel beads were prepared by injecting the mixed emulsions into a 1.5% CaCl_2_ solution (*w*/*v*). The microgel beads with different volume ratios of W_1_/O phase-to-W_2_ phase were marked as Gel-4:6, Gel-5:5, Gel-6:4, Gel-7:3 and Gel-8:2. These were washed 2–3 times with sterile water after 30 min of hardening and stored at 4 °C for 12 h.

### 2.8. Encapsulation Properties and Characterization of Microgel Beads

#### 2.8.1. Viability of Probiotics in Storage Experiments

An amount of 1 g of microgel beads was stored at 4 °C in airtight vials. An amount of 0.1 g of these was removed and dissolved in 9.9 mL of sodium citrate solution (5.0%, *w*/*v*) at 1, 3, 5 and 7 d. *Lactobacillus reuteri* was counted as described above.

#### 2.8.2. Viability of Probiotics in In Vitro Simulated Digestion Experiments

With reference to the INFOGEST model, one simulated gastric fluid (SGF, 2 mg/mL NaCl, 3.3 mg/mL pepsin, 33 µg/mL CaCl_2_, pH adjusted to 2.0 using 1 M HCl) and another simulated gastric fluid (SIF, 35 mg/mL bile salts and 150 mg/mL trypsin dissolved in PBS solution, 160 mg/mL NaCl, 1.32 mg CaCl_2_, pH adjusted to 7.0 with 1 M NaOH) were pre-prepared [20]. All reagents were sterilized by autoclaving or membrane filtration, except for the enzymes used above.

Stomach: A 1 g of microgel beads and pepsin were dissolved in a 9 mL of SGF solution (pH 2) and shaken at 37 °C for 2 h at 100 rpm. An amount of 0.1 g of microspheres were removed at hourly intervals during digestion and dissolved in 9.9 mL of sodium citrate solution (5%, *w*/*v*). The released *Lactobacillus reuteri* were counted by the flat colony counting method. At the end of the digestive phase in the stomach, the characterizations of the microgel beads were recorded by macroscopic images and optical microscope photographs.

Small intestine: After 2 h, a 7.5 mL SIF was rapidly added to the in vitro digestive solution. The pH of the simulation solution was adjusted to 7.0. The operation was as above.

Colon: Finally, the pH of the solution was adjusted to 6.8. The operation was as above. All phases of digestion were protected from light.

### 2.9. Statistical Analysis

All experiments were carried out at least three times. The results were reported as the mean ± standard deviation of triplicate measurements. Statistical significance of differences in means (*p* < 0.05) was tested by one-way analysis of variance (ANOVA) using SPSS 25.0 software (IBM, New York, NY, USA).

## 3. Results and Discussion

### 3.1. Characterization of Zein, AP and ZAHPs

#### 3.1.1. Particle Size, ζ-Potential and PDI

Particle size, ζ-potential and PDI of the hybrid particles are crucial properties that regulate the stability of the emulsions. The ζ-potential reflects the charge on the surface of the particles. The particles have satisfactory homogeneity and stability when the PDI value is below 0.30. Figure 1A shows that the particle size of the zein nanoparticles was about 123.03 nm with a PDI of 0.18. In this experiment, the pH of the dispersion (pH = 4.0) was lower than the isoelectric point of zein (pI = 6.2), resulting in the protonation of the amino group with a positive potential (+30.53 mV). Positively charged zein nanoparticles produced electrostatic repulsive forces that exceeded the hydrophobic forces that could lead to aggregation. This is why zein dispersions were extremely homogeneous in the dispersion system. The particle size of the AP nanoparticles was about 796.40 nm, with a PDI of 0.28 (Figure 1A). As shown in Figure 1B, the ζ-potential of the AP nanoparticles was negatively charged (−26.43 mV). This was attributed to the fact that AP is a complex polysaccharide macromolecule that carried a large number of carboxyl groups. Apparently, with the increase of AP, the particle size of ZAHPs increased sharply to 746.07 nm and then varied at about 600–700 nm. This suggests that AP adsorbed on the surface of zein nanoparticles influenced the self-assembly behavior of the protein [21]. From Figure 1B, it can be seen that the surface of the ZAHPs was negatively charged and that the absolute values of ζ-potential increased with the increase of AP, suggesting an electrostatic interaction between zein and AP. The ζ-potential of the ZAHPs (25.87 mV) was similar to that of AP (26.43 mV) when the mass ratio was reduced to 2:1 (*p* > 0.05), suggesting that AP played a dominant role in the potential through long chain extension to the surface of zein (Figure 1B) [9]. The dispersion of this sample had excellent homogeneity, with a PDI of 0.30. According to DLVO theory, the electrostatic repulsion of the system increases with increasing ζ-potential, which contributes to the increase in energy required to overcome the energy barrier E_max_. Thus, a higher ζ-potential facilitates the formation of a more stable system.

#### 3.1.2. FTIR Spectra

The interaction force between zein and AP is inferred from the vibrations of the different chemical bonds. As can be seen from Figure 1C, the FTIR spectrum of zein revealed several specific bands. The characteristic peaks of C=O stretching vibration at 1650–1660 cm^−1^ (Amide I) were observed. The peaks at 1533–1541 cm^−1^ (Amide II) were assigned to N–H vibrations and C–N stretching vibrations [22]. The spectrum of AP showed broad peak at 3100–3400 cm^−1^ as a result of O–H stretching vibrations, indicating that AP was strongly hydrophilic [23]. The other peaks of AP showed the C=O stretching vibration of the ester bond at 1741 cm^−1^ and the asymmetric stretching vibration of a carboxylate ion band (COO–) at 1616 cm^−1^ [22,24]. Notably, the absorption peaks of ZAHPs were blue-shifted after the addition of AP, suggesting an electrostatic interaction between zein and AP. In addition, the peaks of ZAHPs shifted to 3308–3354 cm^−1^, indicating that the amide group of zein and the hydroxyl or carboxyl group of AP formed new hydrogen bonds [25]. In summary, the formation of ZAHPs was driven by non-covalent bonding interactions. In addition to electrostatic interactions, hydrogen bonding also contributed to the formation of ZAHPs.

#### 3.1.3. Wettability and Microscopic Morphology

Measurement of the *θ* values of the particles can indicate the wettability of the emulsifier. The partial wettability particles adsorbed on mutually immiscible interfaces provide a spatial resistance that prevents the aggregation of droplets. We compared the differences in *θ* of ZAHPs with different mass ratios of zein-to-AP, as shown in Figure 2A. As expected, the zein with the *θ* of 123.27 ± 2.67° seemed to be more readily immersed in the hydrophobic phase.^24^ This indicates that the use of zein alone as an emulsifier was detrimental to the establishment of the stable W_1_/O/W_2_ emulsions [26]. The *θ* values of the ZAHPs were all smaller than that of zein as the addition of AP changed its hydrophobic properties [23]. Moreover, ZAHPs with a *θ* of below and closest to 90° were considered ideal particles for stabilizing the interface of the outer emulsion phase (O/W_2_) when the mass ratio of zein-to-AP was 2:1 (*θ* = 89.88 ± 0.88°). This result provides vital information for the further selection of suitable hybrid particles to stabilize W_1_/O/W_2_ emulsions.

The surface morphologies of the lyophilized nanoparticles can be observed in Figure 2B. Zein nanoparticles were observed as spheres with smooth surfaces and displayed a uniform particle size distribution, which is consistent with the PDI described above. AP nanoparticles were observed as lamellar structures with rough surfaces [27]. Significantly, ZAHPs were seen to be larger in particle size than zein with the addition of AP [18]. The changes in the microscopic morphologies of the hybrid nanoparticles indicate that AP were successfully adsorbed on the surfaces of zein to form “core–shell” structures. The blurred edges between ZAHPs were mainly due to the adhesive properties of AP. We found that the hybrid nanoparticles showed uniform spherical appearances at zein-to-AP mass ratios of 4:1 and 2:1. The results confirm that AP is an effective material with which to enhance the anti-aggregation properties of zein.

### 3.2. Effect of Different Preparation Parameters on the Formation of W_1_/O/W_2_ Emulsions

#### 3.2.1. PGPR Concentration

Figure 3A illustrates the appearances of W_1_/O/W_2_ emulsions. The red lines marked the delamination of freshly fabricated W_1_/O/W_2_ emulsions with concentrations of 0.5% and 1% PGPR [28]. Furthermore, the special multi-chambered structures of W_1_/O/W_2_ were clearly observed. The droplet homogeneity of W_1_/O/W_2_ emulsions prepared at 3% PGPR concentration was significantly improved against those at 0.5%, 1% and 2%. Therefore, to ensure improved densification and stability of the interfacial film, a higher PGPR concentration is required as an emulsifier. However, it was found that excess PGPR would react competitively with the external phase emulsifiers, leading to the formation of aggregates [29]. Similarly, it was observed that the emulsion droplets of 4% PGPR were not homogeneous.

According to Figure 3B, the O phases of the W_1_/O/W_2_ emulsion droplets are labelled in green and zein is labelled in red. The multiple black droplets that appeared in the interior of W_1_/O/W_2_ emulsions are undyed water droplets. The results are consistent with the optical microscope images, indicating the successful preparation of W_1_/O/W_2_ with “droplets within droplets” structures. It is worth noting that the protein circles in which the 4% PGPR emulsion droplets were stained appeared to be discontinuous.

Differences in the physical stability of W_1_/O/W_2_ emulsions with different PGPR concentrations were compared by centrifugation and storage. As shown in Figure 3C, W_1_/O/W_2_ emulsions stabilized with 0.5%, 1% and 2% PGPR concentrations of leaked oil when subjected to external forces. This might be the result of too few surfactants to completely cover the surface of W_1_/O phase. Storage stability is an important property for commercial applications of W_1_/O/W_2_ emulsions. CI could reflect the environmental resistance of W_1_/O/W_2_ emulsions during storage. After 5 d of refrigeration, W_1_/O/W_2_ emulsions showed positive resistance to phase separation at 3% and 4% PGPR concentrations. However, prolonged storage might lead to increased environmental sensitivity of W_1_/O/W_2_ emulsions (Figure 3D). Overall, surfactant concentration was found to be one of the most crucial parameters when determining the physical properties of W_1_/O/W_2_ emulsions. The minimum dose of PGPR that is able to stabilize W_1_/O/W_2_ emulsions would be considered for use in this study.

#### 3.2.2. The Volume Ratios of W_1_/O Phase-to-W_2_ Phase

As shown in Figure 4A, and in contrast with the other simples, the visualized image shows that the W_1_/O/W_2_ emulsions delaminated at a volume ratio of 8:2. The dispersion of droplets was more uniform at a volume ratio of 6:4. With increasing volume ratios of the O phase, W_1_/O/W_2_ emulsions gradually aggregated [30]. Similarly, the microstructures of “droplets within droplets” were successfully observed in W_1_/O/W_2_ emulsions with different volume ratios. The same dense red rings were observed at volume ratios of 5:5 and 6:4 and can be shown in Figure 4B. The hybrid nanoparticles were completely adsorbed on the oil–water interface, providing more spatial site resistance for W_1_/O/W_2_ emulsions. In addition, ZAHPs also avoided the aggregation of droplets due to electrostatic interaction, which is beneficial to the physical properties of W_1_/O/W_2_ emulsions. Clearly, only the W_1_/O/W_2_ emulsion with a volume ratio of 8:2 tended to oil off after centrifugation (Figure 4C). The occurrence of destabilization was attributed to the fact that, as the volume of the W_2_ phase decreased, a smaller number of ZAHPs became insufficient to completely cover the droplet surface [29,31,32,33]. In addition, the increase in the proportion of the O phase might reduce the distances between droplets, ultimately leading to the absence of spatial resistance between droplets [23].

One month later, W_1_/O/W_2_ emulsions showed serious oil leakage at the volume ratios of 7:3 and 8:2 (Figure 4D). With increased storage time, W_1_/O/W_2_ emulsions tended to become unstable, showing, for example, agglomeration, flocculation and Ostwald ripening. However, a sufficient amount of ZAHPs could constitute highly elastic interfacial layer networks to effectively inhibit the aggregation of oil droplets and retard the instability of W_1_/O/W_2_ emulsions [23]. In conclusion, both PGPR concentrations and the volume ratios of W_1_/O phase-to-W_2_ phase were found to strongly influence the environmental resistance of W_1_/O/W_2_ emulsions.

### 3.3. Rheological Property of W_1_/O/W_2_ Emulsions

Increasing the viscosities of W_1_/O/W_2_ emulsions contributes to the prevention of coalescence and flocculation. The effects of preparation parameters on the stability of W_1_/O/W_2_ emulsions were systematically investigated and apparent rheology values of W_1_/O/W_2_ emulsions decreased with increasing shear rate (Figure 5A). It was demonstrated that W_1_/O/W_2_ emulsions exhibited pseudoplastic fluid behaviors. The viscosities of the prepared W_1_/O/W_2_ emulsions increased with the increase in PGPR concentration, which can be attributed to the fact that PGPR reduced the frequency of droplets colliding with each other (Figure 5B) [34,35]. Meanwhile, the storage modulus (*G′*) and loss modulus (*G″*) of W_1_/O/W_2_ emulsions showed a slight frequency dependence on the sweep range of 0.1–10 Hz in the frequency sweep test. Clearly, W_1_/O/W_2_ emulsions were dominated by elasticity. Therefore, they showed solid-like behavior (*G′* > *G″*). The three-dimensional network entanglement properties of ZAHPs at the interfaces of the W_1_/O/W_2_ emulsions were disrupted at increasing frequencies, resulting in a change in colloidal properties to a liquid behavior. In particular, W_1_/O/W_2_ emulsions prepared with 3% PGPR concentration exhibited better viscoelasticity.

W_1_/O/W_2_ emulsions with volume ratios of 4:6, 5:5, and 6:4 were found to be greater in mechanical rigidities (Figure 5B). One possible reason for this was that the proportion of AP in W_1_/O/W_2_ emulsions gradually decreased as the volume ratio of the W_1_/O phase increased. Firstly, AP as a thickening agent required a large amount of space in the rotation of the emulsions system and the intermolecular friction increased with a higher frequency of intermolecular collisions, leading to the enhanced viscosity of the system [24]. Previous studies have shown that the addition of pectin results in shorter emulsion gel formation times and greater independence and hardness [26]. Another major reason for the reduced viscoelasticity of emulsions was that too high a volume ratio of W_1_/O phase results in a less efficient rearrangement of ZAHPs at the interface, which then fails to form a more ordered network structure.

### 3.4. Stability and Morphology of “Egg-Box” Shaped Microgel Beads

#### 3.4.1. Storage Stability of Microgel Beads

As shown in Figure 6, we compared the effects of different volume ratios of W_1_/O phase to W_2_ phase on the storage stability of the microgel beads. It became clear that the number of living cells in all microgel beads loaded with probiotics gradually decreased when stimulated by water, oxygen and temperature. After 7 d, the storage viabilities of Gel-6:4, Gel-7:3 and Gel-8:2 decreased to 6.44, 6.52 and 6.26 log CFU/g, respectively (Figure 6A). Correspondingly, the survival rates for storage in the three groups mentioned above were 74.94%, 78.46% and 73.79%, respectively. Gao et al. have reported that probiotics prepared as microbeads, by exposing them to a separate lipid environment, were dried and, after 4 months at 4 °C, the number of surviving probiotics of Gel-Mb, Gel-Pec-Mb, and Cro-Pec-Mb were 6.87, 6.76 and 6.74 log CFU/g, respectively [27]. Their results demonstrate that drying and lipid environment effectively block stimuli such as oxygen, moisture, etc. The gel layer of microgel beads, ZAHPs, and the O phase provided a physical barrier from environmental factors, such as water and oxygen, for the internal probiotics. However, the numbers of surviving *Lactobacillus reuteri* in Gel-4:6 and Gel-5:5 were found to be less than 4.00 log CFU/g. The results further illustrate that the stability of the internally encapsulated W_1_/O/W_2_ emulsion seemed to play an essential role in the environmental sensitivity of the whole gel delivery system. Improving the stability of the emulsions was the basic strategy for colonic targeting and efficient release of synbiotics.

#### 3.4.2. Digestive Stability of Microgel Beads

Ensuring the release, adsorption and colonization of adequate amounts of probiotics at the target site is the ultimate goal of the delivery system [9]. In this experiment, *Lactobacillus reuteri* and FOS were encapsulated in W_1_ phase. A concentration of 3% PGPR concentration was selected for the preparation of microgel beads based on the effect of surfactant on the stability of W_1_/O/W_2_ emulsions. The in vitro digestion experiment consisted of three phases: stomach, small intestinal and colon. It was evident that the freshly prepared microgel beads were similar in appearance to eggs due to the white color of the zein dispersion after rotary evaporation (Figure 7A). In addition, the network gel layer formed by PLL-alginate-Ca was tightly wrapped, forming a box on the surface of multiple W_1_/O/W_2_ emulsions droplets. Next, on the basis of the W_1_/O/W_2_ emulsions prepared at different volume ratios, we compared the differences in the protective abilities of the microgel beads against *Lactobacillus reuteri* in each group.

After a period of digestion, some oil droplets could be observed floating in the upper layer of the simulated digestive solution. The microgel beads suffered swelling and disruption at neutral pH condition, leading to the breakdown of the oils escaping from the internal emulsion by bile salts and pancreatic enzymes, and the formation of a turbid system (Figure 7B). As shown in Figure 7C, before the experiment, the cell survival numbers of *Lactobacillus reuteri* in all groups were more than 8.00 log CFU/g. After 1 h, the number of surviving cells of free *Lactobacillus reuteri* decreased from 8.95 to 7.02 log CFU/g. At the end of the simulated digestion in the stomach, the cell survival number of the free group was as low as the limit value, indicating that *Lactobacillus reuteir* was sensitive to gastric acid. Conversely, the numbers of surviving cells in Gel-4:6, Gel-5:5, Gel-6:4, Gel-7:3 and Gel-8:2 were 8.53, 8.17, 7.88, 8.14 and 7.82 log CFU/g, respectively. The “box” layer was not vulnerable to degradation by gastric acid and protease stimulation, demonstrating that the three-dimensional network gel coating of the microgel beads effectively improved the stability of the delivery system. Gel-4:6, Gel-5:5 and Gel-6:4 showed excellent protection of probiotics even after the end of simulated digestion in the small intestine. Surprisingly, at the end of the colonic digestion phase, the numbers of live probiotics in the three groups above were still within the recommended number (>6 log CFU/g). In Figure 7D, Gel-6:4 had the highest final survival rate of 75.11% compared with the other groups. The dotted lines in Figure 7D show that probiotic survival rates were lower than the corresponding values but that their exact values could not be observed through the plate counting method.

These results indicate that the “egg-box” shaped microgel beads were resistant to digestive enzymes and bile salts. Firstly, a higher volume ratio of W_1_/O phase helped to increase the internal droplet space, thus encapsulating a larger number of probiotics. Meanwhile, the oil-in-water interface required sufficiently dense ZAHPs as a robust and ordered physical barrier by which to prevent direct contact between components of the digestive tract and probiotics, such as digestive enzymes and H^+^ [27,36]. Finally, as the porous networks of calcium alginate were filled by the added PLL, dense three-dimensional gel structures as barriers were further formed. In this study, we established a pH-sensitive and enzyme-triggered colon-targeted delivery system loaded with synbiotics. In the stomach, the “box” shaped gel shell of PLL-alginate-Ca was resistant to H^+^ and proteases. AP adsorbed on the zein surface also effectively reduced the aggregation of W_1_/O/W_2_ emulsions induced by proteases and ionic concentrations through steric repulsion. In the small intestine, enhanced electrostatic repulsion of the anionic groups due to deprotonation of the carboxyl groups (COO-) of calcium alginate and OH^−^ infiltration reacting with Ca^2+^ both resulted in the disintegration of the partially cross-linked network structure during the digestive phase [37,38]. At the same time, the partially disintegrated gel shell released W_1_/O/W_2_ emulsions inside. The desorption of AP from the surface of zein was attributed to electrostatic repulsion, which facilitated the layer-by-layer decomposition of W_1_/O/W_2_ emulsions [9]. W_1_/O/W_2_ emulsions would form larger aggregates leading to a significant increase in particle size and destabilization. AP as a prebiotic would be degraded by colon-specific enzymes, leading to further disintegration of the delivery system. All probiotics were released in order to work synergistically with prebiotics to maintain intestinal balance. Therefore, the dual-responsive microgel beads of this study are appropriate colon-targeted carriers for the delivery of synbiotics.

## 4. Conclusions

In this study, we selected ZAHPs with a suitable contact angle to successfully fabricate W_1_/O/W_2_ emulsions with a special multi-chambered structure. W_1_/O/W_2_ emulsions prepared with 3% PGPR concentration and volume ratio of W_1_/O phase-to-W_2_ phase of 6:4 provided satisfactory stability. Based on this, dual-responsive “egg-box” shaped microgel beads loaded with synbiotics were successfully prepared. In vitro digestion simulations showed that the microgel beads provided bidirectional physical barriers that contributed to the sustained and effective release of probiotics. On the one hand, it prevented the escape of internal probiotics, while, on the other, it avoided the erosion of external and harsh factors. Overall, this study provides promising insights into an intelligent responsive colon-targeted delivery system for bioactives, and the biological fate and physiological effects of probiotics will be explored in our future studies.

## Figures and Tables

**Figure 1 foods-13-02163-f001:**
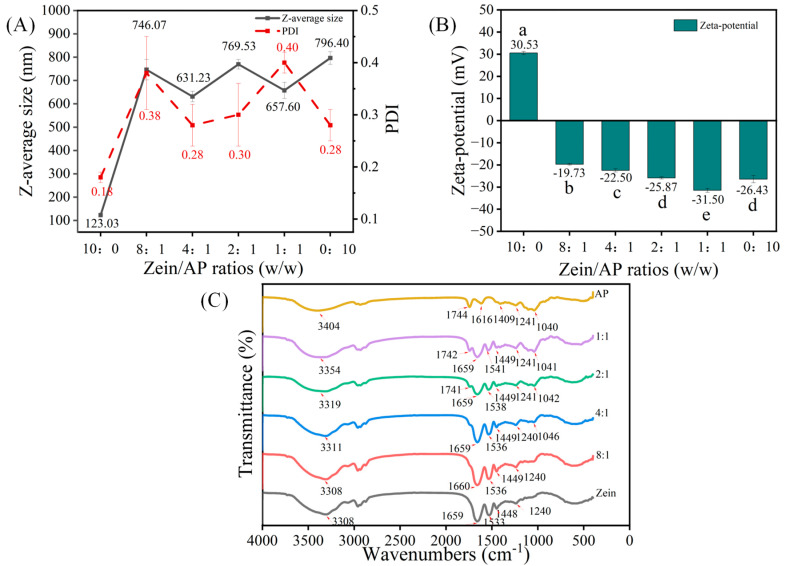
(**A**) Particle size, PDI of zein, AP and ZAHPs; (**B**) ζ-potential of zein, AP and ZAHPs; (**C**) FTIR spectra of zein, AP and ZAHPs. Different lower-case letters on the columns represent significant differences (*p* < 0.05).

**Figure 2 foods-13-02163-f002:**
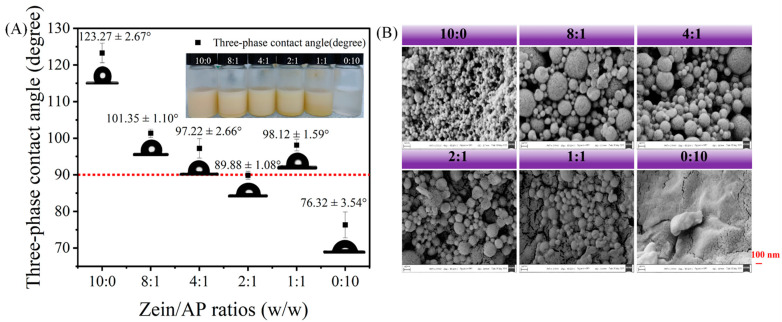
(**A**) The *θ* values of zein, AP and ZAHPs and (**B**) the microstructures of freeze-dried zein, AP and ZAHPs under scanning electron microscope.

**Figure 3 foods-13-02163-f003:**
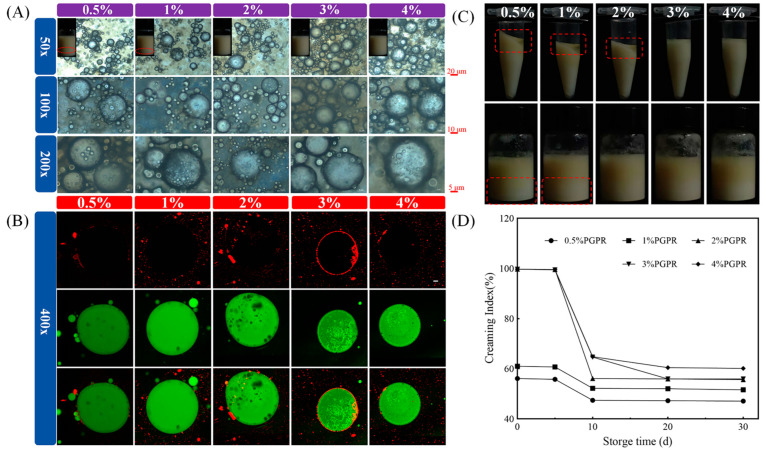
Structures and physical stabilities of W_1_/O/W_2_ emulsions prepared with different PGPR concentrations (0.5%, 1%, 2%, 3% and 4%, *w*/*v*). (**A**) Visual appearances and microstructures of freshly prepared W_1_/O/W_2_ emulsions; (**B**) microstructures of stained W_1_/O/W_2_ emulsions under confocal scanning electron microscopy. Scale bar = 2 μm. (**C**) Visual observations of fresh W_1_/O/W_2_ emulsions after centrifugation and (**D**) visual photographs of W_1_/O/W_2_ emulsions during storage (10, 20 and 30 d).

**Figure 4 foods-13-02163-f004:**
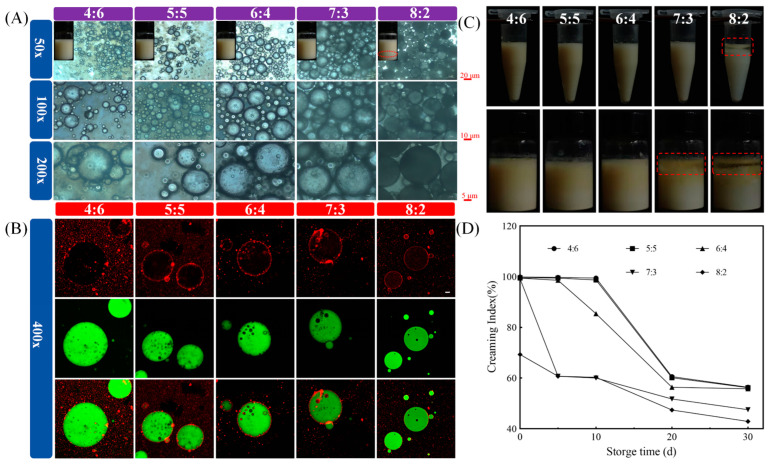
Structures and physical stabilities of W_1_/O/W_2_ emulsions with different volume ratios of W_1_/O phase to W_2_ phase. (**A**) Visual appearances and microstructures of freshly prepared W_1_/O/W_2_ emulsions, (**B**) microstructures of stained W_1_/O/W_2_ emulsions under confocal scanning electron microscopy, (**C**) visual observations of fresh W_1_/O/W_2_ emulsions after centrifugation, and (**D**) visual photographs of W_1_/O/W_2_ emulsions during storage (10, 20 and 30 d).

**Figure 5 foods-13-02163-f005:**
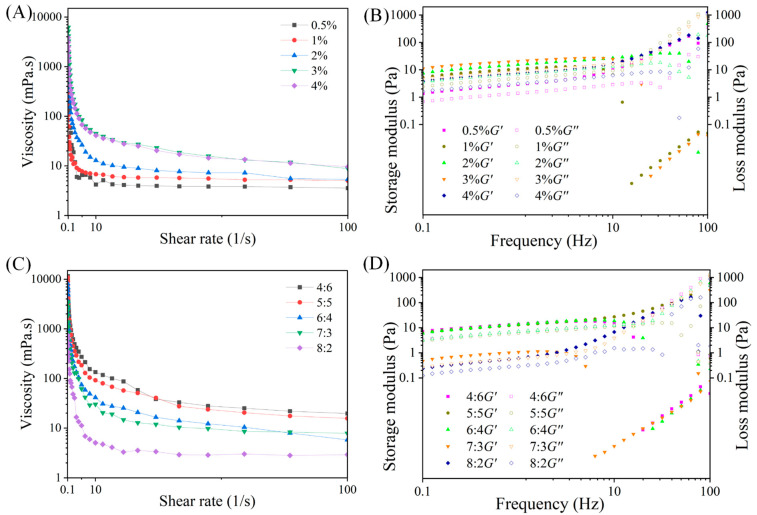
(**A**) and (**B**) show the apparent viscosity curves and frequency sweep curves with different PGPR concentrations, while (**C**) and (**D**) show the apparent viscosity curves and frequency sweep curves with different volume ratios of W_1_/O phase-to-W_2_ phase.

**Figure 6 foods-13-02163-f006:**
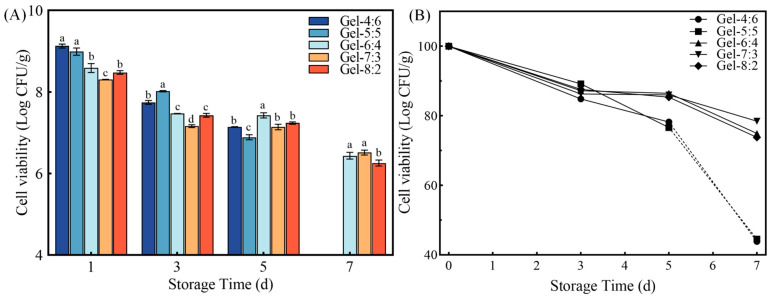
Effects of “egg-box” shaped microgel beads on the storage viability of *Lactobacillus reuteir*. (**A**) The numbers of viable cells of *Lactobacillus reuteri* and (**B**) the survival rates of *Lactobacillus reuteri*. Different lower-case letters on the columns represent significant differences (*p* < 0.05).

**Figure 7 foods-13-02163-f007:**
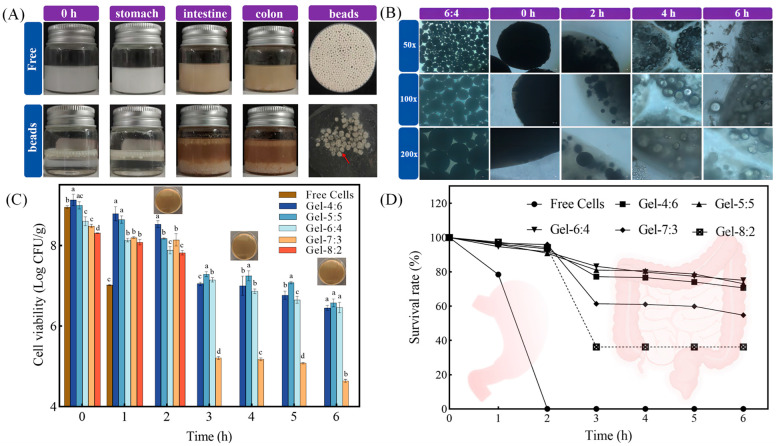
Behaviors of “egg-box” shaped microgel beads during in vitro simulated digestion. (**A**) Visual images of pre-digested, stomach, small intestine and colon; (**B**) microstructures of fresh, stomach, small intestine and colon; (**C**) the numbers of viable cells of *Lactobacillus reuteri*; and (**D**) the survival rates of *Lactobacillus reuteri*. Different lower-case letters on the columns represent significant differences (*p* < 0.05).

## Data Availability

The original contributions presented in the study are included in the article, further inquiries can be directed to the corresponding author.

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
