# Peer review of "Dual-Responsive “Egg-Box” Shaped Microgel Beads Based on W1/O/W2 Double Emulsions for Colon-Targeted Delivery of Synbiotics"

_foods, 2024, doi:10.3390/foods13142163_

Round 1

Reviewer 1 Report

Comments and Suggestions for Authors

The scientific and research topic addressed by the study it is current and of interest to the scientific community in developing solutions that targets bioactive food for human health. The research is well planned and conducted in line with the scope of the research, with appropriate methodology and adequately selection of analysis methods.

The originality elements of the study consists in the "egg-box" design of delivery systems for probiotics and the new composition/mixture formulations of the inner and outer layers of the multi-chambered emulsion-based system. The research approach targets an important solution in the field namely to enhance the resistance of natively susceptible probiotics to environmental factors by incorporating the emulsions into protective microgel beads and also to improve storage stability. Moreover, the integration of pH-sensitive and enzyme-triggered materials into the "egg-box" system ensures a controlled and target release of the probiotics at colonic site.

The complex approach of stabilizing both the inner content by using zein-apple pectin hybrid nanoparticles and external protection by developing a resistant and pH-responsive poly-L-lysine (PLL)-alginate-CaCl2 (Ca) cross-linked gel represent an innovative aspect. The conclusions of the article are fine correlated with the experimental results, however, a possible improvement can be considered in terms of including other specific particularities of the prepared emulsions (e.g. rheological behavior).

Some minor observations are listed below:

-           Avoid using abbreviation if otherwise firstly defined (e.g. abstract - AP)

-           The introduction could be better organized and improved the gradation of information. The authors should avoid extensive description of the research work in the introduction section. The concept and research approach should be more concisely presented. Lines 66-67: should be carefully checked and rephrased and also lines 76-77 are unclear and should better expressed.

Reviewer 2 Report

Comments and Suggestions for Authors

The authors present an approach for encapsulating Lactobacillus plantarum (LP) in W1/O/W2 double emulsions to enhance its viability and targeted delivery during simulated gastrointestinal conditions. The study is well-designed and the results are promising. However, I have some concerns that need to be addressed before the manuscript can be considered.

The authors state that "The outer aqueous phase (W2 phase) and the intermediate oil phase (O phase) provided multi-layered physical barriers for probiotics avoiding directly contact with H+, digestive enzymes and bile salts" (Line 46). To further strengthen this claim, I suggest including a reference to another relevant study, such as: An Updated Comprehensive Overview of Different Food Applications of W1/O/W2 and O1/W/O2 Double Emulsions (10.3390/foods13030485)

I noticed that the authors have cited several of their own previous works throughout the manuscript. While self-citation is sometimes appropriate, it should be done sparingly and only when the cited work is directly relevant to the current study. In some cases, I found that the cited work did not fully support the claims made in the manuscript. For example: The authors state that "However, W1/O/W2 emulsions are susceptible to flocculation, coalescence and Ostwald ripening under the influence of mechanical stress, thermal energy, time and pH" and cite their own publications [15, 16] in support of this claim. However, I reviewed these publications and did not find that they provide sufficient evidence to fully support this statement. I suggest that the authors provide additional evidence and remove the self-citations.

In addition, I found that the authors self-cited their work [19] even though the method used to prepare the emulsions in the current study (stirring) was different from the method used in [19] (ultrasound-assisted). Therefore, I do not recommend citing [19] in the current manuscript.

This manuscript describes the development of "egg-box" shaped microgel beads for the delivery of synbiotics (a combination of probiotics and prebiotics) to the colon. The approach utilizes W1/O/W2 double emulsions stabilized with zein-apple pectin hybrid nanoparticles (ZAHPs) and coated with a poly-L-lysine (PLL)-alginate-CaCl2 (Ca) crosslinked gel layer. The study is well-structured and presents a detailed methodology for the fabrication and characterization of the microgel beads. However, there are some critical aspects that require further clarification or justification before publication.

The authors refer to the microgel beads as "egg-box" shaped. However, based on the provided descriptions and images, the shape appears more spherical or semi-spherical. It is recommended to revise the terminology to accurately reflect the actual morphology of the microgel beads.

In Figure 1B, the legend "Different lower-case letters on the columns represent significant differences (p<0.05)."  is awkwardly positioned

The manuscript mentions the presence of a peak at 1616 cm-1 in the FTIR spectrum of apple pectin (AP), which is attributed to the asymmetric stretching vibration of the free carboxyl group (COO-), it strikes me how you differentiate between RCOOR and RCOOH, please explain in brief. Anyhow, there is no discussion on how this peak interacts with other components or how it changes in the ZAHP formulations. Clarification on this point is necessary.

The particle size distribution obtained by dynamic light scattering (DLS) in Figure 1A appears inconsistent with the scanning electron microscopy (SEM) images in Figure 2B. The SEM image shows a broader size distribution, with some particles reaching around 100 nm, particularly for the 8:1 ratio. This discrepancy needs to be addressed. The authors should explain the potential reasons for the observed difference and ensure consistency between the characterization techniques.

The manuscript states that a higher concentration of PGPR (emulsifier) is needed for improved interfacial film densification and stability. However, it also mentions that excess PGPR can react competitively with the external phase emulsifiers, leading to aggregate formation. This contradiction requires further explanation.

The study reports oil leakage from W1/O/W2 emulsions at specific volume ratios (7:3 and 8:2) after one month of storage. Additionally, the emulsions exhibited signs of instability, such as agglomeration, flocculation, and Ostwald ripening. While the authors mention that ZAHPs can improve emulsion stability, the provided data suggests limitations in their effectiveness.  These stability issues should be addressed in more detail. The authors should explore strategies to enhance the long-term stability of the emulsions.

The results indicate that the number of surviving Lactobacillus reuteri in some gel formulations (Gel-4:6 and Gel-5:5) was lower than expected. This suggests that the stability of the encapsulated W1/O/W2 emulsions might be compromising the overall efficacy of the delivery system. The authors should delve deeper into this observation and propose potential solutions to improve the survival rate of probiotics within the microgel beads.

L. plantarum cells are rod-shaped with dimensions of 0.9–1.2 μm wide and 3–8 μm long. Given the reported sizes of the emulsions and microgel beads, it is difficult to comprehend how these rod-shaped bacteria can be effectively encapsulated within these structures. Please provide a convincing explanation for this seemingly contradictory aspect. The reported sizes of the emulsions and microgel beads are considerably smaller than the dimensions of L. plantarum cells. This raises questions about the feasibility of encapsulating individual bacterial cells within these structures. Please provide high-resolution microscopy images that clearly demonstrate the encapsulation of individual L. plantarum cells within the emulsions and microgel beads.

Reviewer 3 Report

Comments and Suggestions for Authors

This manuscript presents the development of an advanced, intelligent colon-targeted delivery system for synbiotics. The system features microgel beads with an “egg-box” configuration. These beads are composed of W1-O-W2 double emulsions, stabilized by zein/apple pectin particles forming the inner "egg," and an outer shell of poly-L-lysine-alginate-CaCl2 constituting the "box." This specific design effectively protects the probiotic encapsulated within the inner W1 phase.

The manuscript is well-written, with statements supported by substantial evidentiary material. All synthesis methods, characterization techniques, and applications of the developed systems are meticulously detailed.

I recommend minor to moderate revisions having in mind the following critical notes:

11. The concluding part of the introduction should be edited from lines 74 to 102 2to avoid repetition, clearly indicate the difference from the studies of other authors (e.g. [17]), and the novelty of the present study.

12. Line 70-73: “Our previous study…” – what study, missing citation.

13. The scale bars are not visible, especially in Fig.3B, please correct.

14. Line 291:  Fig 2A instead of 3A

Comments on the Quality of English Language

11. Line 76-77 - editing needed, poor wording

12. Line 428-429 – editing needed, poor wording.

Round 2

Reviewer 2 Report

Comments and Suggestions for Authors

The authors diligently addressed the comments and responded convincingly to the issues raised by this reviewer, so I believe that the quality and contribution of the manuscript has improved.